# Targeting Non-Alcoholic Fatty Liver Disease with Hawthorn Ethanol Extract (HEE): A Comprehensive Examination of Hepatic Lipid Reduction and Gut Microbiota Modulation

**DOI:** 10.3390/nu16091335

**Published:** 2024-04-29

**Authors:** Tianyu Wang, Dawei Wang, Yinghui Ding, He Xu, Yue Sun, Jumin Hou, Yanrong Zhang

**Affiliations:** 1College of Food Science and Engineering, Changchun University, Changchun 130022, China; wangty84@ccu.edu.cn (T.W.); dyh155856799771023@163.com (Y.D.); xh1796293949@163.com (H.X.); 15844448981@163.com (Y.S.); houjm@ccu.edu.cn (J.H.); 2School of Food Science and Engineering, Jilin Agricultural University, Changchun 130118, China; wangdawei@jlau.edu.cn

**Keywords:** nonalcoholic fatty liver disease, hawthorn ethanol extract, gut microbiota, high-fat diet, short-chain fatty acids

## Abstract

Recent studies have highlighted the lipid-lowering ability of hawthorn ethanol extract (HEE) and the role played by gut flora in the efficacy of HEE. Our study sought to explore the effects of HEE on non-alcoholic fatty liver disease (NAFLD) in normal flora and pseudo germ-free mice. The results showed that HEE effectively diminished hepatic lipid accumulation, ameliorated liver function, reduced inflammatory cytokine levels and blood lipid profiles, and regulated blood glucose levels. HEE facilitated triglyceride breakdown, suppressed fatty acid synthesis, and enhanced intestinal health by modulating the diversity of the gut microbiota and the production of short-chain fatty acids in the gut. In addition, HEE apparently helps to increase the presence of beneficial genera of bacteria, thereby influencing the composition of the gut microbiota, and the absence of gut flora affects the efficacy of HEE. These findings reveal the potential of hawthorn for the prevention and treatment of NAFLD and provide new perspectives on the study of functional plants to improve liver health.

## 1. Introduction

The clinicopathologic manifestation of nonalcoholic fatty liver disease (NAFLD) is an atypical accumulation of triglycerides in the hepatocytes without or with a minimal history of alcohol consumption. As of now, there is no specialized medication proven to effectively prevent or manage NAFLD [1]. Hence, studying the underlying mechanisms of NAFLD and identifying potential targets for intervention is crucial in the prevention and management of NAFLD. The disease’s emergence is thought to be multifactorial, with recent studies demonstrating imbalanced gut microbiota playing a role in NAFLD and contributing to its progression by influencing metabolic outputs, the metabolism of sugars and fats, immunological activities, and inflammatory processes [2,3].

A high-fat diet (HFD) induces ecological dysregulation of the gut microbiota and leads to increased intestinal permeability in mice. Gut microbiota imbalance can lead to inflammatory responses and is closely linked to NAFLD and cancer [4,5,6]. Several gut microbiota and their metabolites play a crucial role in maintaining liver and intestinal health, such as short-chain fatty acids (SCFAs) in the gut, which are closely associated with inflammation, as well as glucose and lipid metabolism [7]. As metabolic byproducts of microbes, SCFAs are not only the preferred energy substrates for the intestinal epithelial cells, but are also crucial components in regulating antioxidants and the balance of the gut microbiota [8]. Therefore, the significance of microorganisms that generate SCFAs is increasingly recognized. The adjustment of the proportions of intestinal microbial communities can be optimized through prebiotics provided in food or dietary supplements.

Previous research has indicated that probiotics have diverse effects on the gut microbiota of mice with NAFLD, including improvements in the metabolism of related metabolites such as SCFAs, bile acids, and digestive enzymes. These functional metabolites can restore liver function, blood lipid levels, and cytokine levels in mice with NAFLD [9,10]. Currently, supplementation with probiotics and prebiotics is the mainstay of improving the balance of the gut microbiota, thereby becoming a potential therapeutic direction for the treatment of NAFLD [11,12]. Recent pharmacological studies have indicated that certain food–medicine dual-purpose items can treat NAFLD by modulating the gut microbiota. Saponins found in ginseng have been shown to suppress inflammatory responses in rats with a model of NAFLD, regulate lipid metabolism, and alleviate liver damage [13]. Ganoderma lucidum polysaccharides are capable of modulating bile acid levels and metabolites of the gut microbiota, and ameliorating NAFLD by altering gut microbiota composition [14,15]. These research findings indicate that herbs act as prebiotics in the gastrointestinal tract when consumed as dietary supplements.

The fruit of the hawthorn (Crataegus pinnatifida), which is extensively utilized in traditional herbal practices for lipid reduction, has been the focus of recent research. Studies indicate that extracts enriched with hawthorn fruit polyphenols can counteract dysfunction in pancreatic β-cells due to high fructose intake by alleviating oxidative stress in the endoplasmic reticulum. The active ingredients include compounds from flavonoids and triterpenes [16]. As a functional food, its extract also protects the vascular endothelium, restores endothelial dysfunction, and promotes endothelial relaxation [17]. Furthermore, hawthorn extract can modulate immune responses and regulate lymphocyte subsets. Additionally, studies have shown that hawthorn extract has a protective effect against microcystin-induced liver injury [18].

Although the abundant flavonoid compounds in extracts have a wide range of pharmacological applications, their poor water solubility often results in low oral bioavailability [19,20]. This indicates that the intake of these extracts through gavage might engage the gut microbiota, subsequently promoting health improvements. The exact mechanism of hawthorn extracts’ effect on lipid metabolism by altering the gut microbiota is still to be fully understood. In this study, we investigated the therapeutic effect of HEE on NAFLD in mice and the contribution of gut microbiota to the regulation of lipid metabolism in HEE by using a HFD that induces NAFLD and a pseudo-bacteria-free mouse model caused by antibiotics, thus laying the foundation for future clinical studies.

## 2. Materials and Methods

### 2.1. Preparation of Hawthorn Ethanol Extract (HEE)

Initially, fresh hawthorn fruits were carefully broken and ground before undergoing drying at a controlled temperature of 45 °C until a fine powder was obtained. Subsequently, the dried powder was subjected to extraction with water at 50 °C, followed by concentration and another drying process. To further refine the extract, it underwent an additional three rounds of extraction using 95% ethanol, resulting in the final product known as HEE. The main active ingredients were detected by HPLC-MS as organic acids and flavonoids, with 252.4 μg of citric acid, 61.5 μg of hawthorn acid, 79.3 μg of caffeic acid, 1,225 μg of catechins, 145.5 μg of quercetin, 34.5 μg of hyperoside, and 14.2 μg of apigenin8-C-glucoside per gram of fresh fruit.

### 2.2. Animal Experiments

Male 5-week-old SPF-grade C57BL/6/J mice were purchased from Vital River Laboratories located in Beijing, China. Mice were randomized into normal dietary control groups after the acclimatization period. After an acclimatization period, mice were randomly divided into a normal-flora control group (NFNDC, TP 23300, Trophic Animal Feed High-Tech Co., Ltd., Nantong, China. The composition of high-fat feed is shown in Appendix A) and four high-fat diet model groups (NFHFD, normal flora model. NFHFDHEE, normal flora HEE. PGHFD, pseudo germ-free model. PGHFDHEE, pseudo germ-free HEE. 60% fat diet TP 23302, Trophic Animal Feed High-Tech Co., Ltd., China) of six mice each and fed for 16 weeks. The dose of HEE was set at 150 mg/kg, a dosage proven to have minimal side effects, thus selected as the optimal dose for the experiment. HEE was dissolved in 0.5% carboxymethyl cellulose (CMC; National Medicines Corporation Chemical Reagent Co., Ltd., Shanghai, China) to prepare the gavage solution. Two of the high-fat diet groups (PGHFD and PGHFDHEE groups) underwent antibiotic treatment for two weeks, during which the PGHFDHEE group and NFHFDHEE group were orally administered 150 mg/kg HEE daily for 4 consecutive weeks, following the antibiotic treatment. The other two groups of mice (NFHFD group and PGHFD group) and the NFNDC group received an equivalent dose of 0.5% CMC. Food consumption, body weight, and fasting blood glucose levels were measured weekly. At week 16, the mice were euthanized, and mouse feces were collected the day before. In a controlled environment, mouse blood was drawn under dim light conditions and allowed to clot for 60 min at 25 °C. The blood was then centrifuged at 3000× *g* for 10 min. Subsequently, the serum was separated and stored in a deep-freeze reservoir at a temperature of −80 °C for subsequent analyses. Liver and epididymal fat weights were recorded. Fecal and tissue samples were stored at −80 °C for further analysis. All animal experimental protocols in this study were reviewed by the Animal Ethics Committee and accurately implemented during use, and the entire experimental process was carried out with animal ethics in mind.

### 2.3. Pseudo-Germfree Mice (PG) Treated with Antibiotics

After a 10-week period on an HFD, two groups of mice from the HFD regimen (PGHFD and PGHFDHEE) were exposed to antibiotic therapy involving a combination of vancomycin (3.125 g), ampicillin (6.25 g), metronidazole (6.25 g), and neomycin (6.25 g) dissolved in 500 mL of saline, delivered via gavage at a dosage of 10 mg per mouse daily [21]. Throughout this intervention, changes in body weight and food consumption were carefully monitored, and fecal specimens were obtained. Following a two-week period post-antibiotic treatment, the PGHFDHEE and NFHFDHEE groups were subjected to a 4-week intervention with HEE extract.

### 2.4. Histopathological Analysis of Animals

Some fresh epididymal fat was fixed by a 4% solution of paraformaldehyde with phosphate-buffered saline, maintained at 4 °C for a period ranging from 2 to 4 h, before being encased in paraffin, after which it was cut into 10 μm sections. The sections were then subjected to staining procedures using hematoxylin and eosin (H&E) for histological examination. Liver tissues were immediately frozen in liquid nitrogen for further oil red O staining. These cryosections were then cut into 10-micron slices at low temperatures and stained with oil red O to assess lipid accumulation. The imagery of the stained tissue sections was captured employing a digital slide scanning system, and the subsequent analysis was conducted using ImagePro Plus v6.0 software to quantify and evaluate the staining results.

### 2.5. Mouse Serum Indicator Assay

The concentrations of serum biochemical markers, including TG, TC, LDL-c, HDL-c, ALT, and AST, alkaline phosphatase (ALP), and cholinesterase (CHE), were quantified utilizing an automated biochemical analysis device (Mindray BS-480; Mindray, Shenzhen, China). Serum cytokines were obtained according to the appropriate kit instructions.

Serum levels of TNF-α, IL-6, LBP, and LPS were measured using ELISA kits (SenBeiJia Biological Technology Co., Ltd., Nanjing, China) according to the kit instructions.

### 2.6. Quantitative Reverse Transcription–Polymerase Chain Reaction (qRT-PCR) Analysis

Total RNA was extracted from colon and liver tissues using TRIzol reagent (Invitrogen, Carlsbad, CA, USA). Subsequent to the isolation, RNA integrity was assessed, ensuring that each RNA sample (1 µg) met the requisite purity and concentration criteria for further processing. Then, these samples were reverse-transcribed into complementary DNA (cDNA) using HiScript III-RT SuperMix (+gDNA wiper; Vazyme, Nanjing, China). For a qualitative assessment of gene expression, the polymerase chain reaction (PCR) was employed alongside agarose gel electrophoresis. This step provided a visual confirmation of the specific amplification and integrity of the target transcripts. The gel electrophoresis enabled the verification of the expected size of PCR products, serving as a preliminary validation of the amplification specificity and efficiency.

Advancing to quantitative analysis, quantitative PCR (qPCR) was executed on a CFX Connect real-time PCR detection system (Bio-Rad, Hercules, CA, USA). The qPCR assay was optimized to ensure specificity, efficiency, and reproducibility, employing SYBR Green or TaqMan probes for the detection of the amplification products. Gene expression was quantified using the 2^−ΔΔCt^ method, thus providing a relative expression level of the gene of interest. Primer sequences for qPCR are shown in Appendix A.

### 2.7. Gut Microbiota Analysis

The MP Stool Genomic DNA Extraction Kit was used to extract total DNA from mouse feces. PCR amplification of the 16S rDNA V3-V4 heavy region was performed according to the instructions, in 50 μL of reaction mixture (including 25 μL of 2× TaqMasterMix solution, 1 μL of each primer, 1 μL of template, and 22 μL of ddH_2_O (BasalMedia Co., Ltd., Shanghai, China)), using this DNA as a template with the universal primer 341F/806R. The PCR products were purified using nucleic acid gel electrophoresis and a DNA Gel Purification Kit. The concentration of purified products was measured with the Qubit™ dsDNA BR Assay Kit, and libraries were prepared using the TruSeq DNA LT Sample Preparation Kit for sequencing on the Illumina MiSeq system. Data analysis was conducted using the QIIME2 platform with the DADA2 plugin, and by annotating species using the q2-feature-classifier plugin with the Silva database reference. Microbial diversity was assessed using the q2-diversity plugin, with alpha diversity indicated by observed OTUs and Faith’s PD indices, and beta diversity analyzed through PCoA based on Bray–Curtis distances among samples [22,23] (all the kits were purchased from Enzyme Exemption Industry Co., Ltd., Yancheng, Jiangsu, China).

### 2.8. Analysis of Short-Chain Fatty Acids

Intestinal content samples were freeze-dried to eliminate moisture. Approximately 50 mg of the dried material was mixed with 500 μL of saturated NaCl and 20 μL of 15.84 mmol/L 2-ethylbutyric acid, shaken after a 30 min soak for full dissolution. The mixture was then acidified with 40 μL of 10% sulfuric acid and vortexed for even blending, followed by the addition of 1 mL pre-chilled ether, and vortexed again. After centrifugation at 10,000× *g* at 4 °C for 15 min, the supernatant was transferred to a new tube. Moisture was removed with 0.25 g of anhydrous sodium sulfate, and the sample was centrifuged again. The aliquot of the clear supernatant was prepared for GC-MS analysis. Samples were analyzed using a GC-MS system (QP2010 Ultra, SHIMADZU, Kyoto, Japan) with an Rtx-Wax column. The setup involved an inlet temperature of 240 °C, an oven starting at 100 °C, helium as the carrier gas, and specific pressures and flow rates to ensure accurate readings. The temperature was gradually increased from 100 °C to 200 °C following a specific program. Various SCFAs were detected using a selected ion monitoring mode in the mass spectrometer, set between 2 and 100 (*m*/*z*) [24].

### 2.9. Statistical Analysis

The data were reported as mean values accompanied by standard deviations. An analysis of the data obtained from the animal experiments was conducted utilizing one-way analysis of variance (ANOVA). Post hoc multiple comparisons among the groups were carried out employing Dunnett’s test. In all statistical analyses, a significance level with a confidence interval of 95% or *p* < 0.05 was used as the critical value for determining statistical significance.

## 3. Results

### 3.1. The Mitigating Effect of HEE on HFD-Induced NAFLD in Mice

We evaluated the impact of HEE on mice suffering from NAFLD induced by HFD. The daily food intake, body weight, liver mass, and blood glucose levels of five mouse groups were monitored (Figure 1). No significant pattern changes were observed in food consumption among the groups (Figure 1B). However, both body and liver weights decreased after HEE treatment, regardless of whether in a normal microbiota environment or under antibiotic treatment (Figure 1C,F). By comparing the weight gain, it can be seen that the model mice gained significantly more weight than the control mice, and in the HEE intervention, mice gained significantly less weight than the model mice (Figure 1C,D). Similar ameliorative effects were observed for the liver-to-body weight ratio and fasting blood glucose levels (Figure 1G,H). Nonetheless, the regulatory effects were not significant in the antibiotic-treated groups, predicting that the disruption of the intestinal flora by antibiotics might have affected the efficacy of HEE.

Figure 2A,B show sections of the liver with oil red-O staining and epididymal fat H&E, which indicated that lipid accumulation in the liver and epididymal fat were significantly reduced in the HEE group compared with the non-HEE group. Serum biochemical markers indicated that HEE improved liver function, significantly decreased ALT and AST levels in the NFHFD and PGHFD groups, and helped to regulate alkaline phosphatase (ALP) levels in the NFHFD group, although it did not significantly improve cholinesterase (CHE) levels (Figure 2C–F). Therefore, HEE effectively modulates lipid accumulation and liver function in HFD mice.

### 3.2. Modulation of Lipid Profile and Inflammation-Related Cytokines by HEE in HFD Mice

A high-fat diet causes dyslipidemia and inflammatory responses in the organism, so we further investigated the modulatory effects of HEE on serum lipids and inflammation-associated cytokines in HFD mice. Our findings reveal that, in comparison to the high-fat diet group, HEE significantly reduces the levels of serum TG, TC, and LDL-c, irrespective of the presence of a normal microbiota or an antibiotic-treated environment. Additionally, HEE notably increased the level of HDL-c in the NFHFDHEE group. This indicates HEE’s potential in modulating lipid profiles and inflammatory responses in the context of dietary-induced hyperlipidemia (Figure 3A–D).

Systemic inflammation in NAFLD mice and the therapeutic effect of HEE were investigated by detecting cytokines in serum (Figure 3E–H). We observed that HEE effectively modulated the levels of tumor necrosis factors α (TNFα), endotoxin lipopolysaccharide (LPS), and lipopolysaccharide binding protein (LBP) in the NFHFDHEE group, while significantly alleviating the levels of LPS and interleukin-6 (IL-6) in the PGHFD group. Consequently, HEE has shown its beneficial effects by reducing the inflammatory response induced by a high-fat diet, thereby decreasing the liver’s susceptibility to the effects of endotoxins. This underscores HEE’s therapeutic potential in ameliorating diet-induced hepatic inflammation.

### 3.3. Effects of HEE on the Expression of Genes Related to Intestinal SCFAs and Hepatic Lipid Metabolism in Mice

Studies indicated that HFD alters the gut microbiome in the cecum and colon, increasing concentrations of SCFAs, such as propionate in the cecum and isovalerate and valerate in the colon. However, the concentration of SCFAs dropped significantly after antibiotic treatment, highlighting the crucial role of the gut microbiome (Figure 4A,C).

Through significant changes in liver and adipose tissue slices and glycerol triacylglycerol indicators, we further investigated the metabolic status of triglycerides in the liver. We selected key enzymes in the lipolytic process, such as adipose triglyceride lipase (ATGL), hormone-sensitive lipase (HSL), and monoacylglycerol lipase (MGL) in adipose tissue. Our results showed that ATGL and HSL gene expression was significantly activated in the NFHFDHEE group compared with the high-fat model group, whereas no significant changes were observed in the PG group. Interestingly, HEE similarly affected the activation of MGL in the PGHFDHEE group, promoting lipolysis (Figure 4B,D–F). Thus, HEE promotes lipolysis in the liver and plays an important role in regulating hepatic lipid metabolism.

### 3.4. HEE Modulates the Gut Microbiota in HFD Mice

To investigate the modulatory effects of HEE on the gut microbiota of high-fat-diet-fed mice, we assessed changes in fecal microbial community structure after the administration of HEE using 16S rRNA analysis. The treatment notably adjusted the prevalence of Bacteroidetes and Firmicutes within the NFHFDHEE group, elevating *Bacteroidetes* quantities to similar levels as those observed in the control group, as illustrated in Figure 5A,B. This adjustment is further reflected by the *Bacteroidetes*-to-*Firmicutes* ratio. Among the antibiotic-treated cohort, *Verrucomicrobia* dominance was observed, which was marked by reduced microbial diversity and lower abundance.

We further delved into the alterations at the genus level within the gut microbiota, as depicted in Figure 5C, showcasing the proportional distribution of dominant bacterial genera across five groups. The gut microbiota composition of SPF-maintained normal mice starkly contrasted with that of pseudo-germ-free mice treated with antibiotics (PG). Notably, in the NFHFDHEE group, genera such as unidentified *Lachnospiraceae*, *Lactobacillus*, the *[Eubacterium] fissicatena group*, *Ruminiclostridium 9*, and *Romboutsia* predominated, suggesting their pivotal physiological roles—a trend similarly observed in the NFNDC group. Conversely, the PG group harbored various potentially pathogenic genera like *Streptococcus*, *Stenotrophomonas*, *Enterobacter*, *Enterobacteriaceae*, and *Escherichia-Shigella*. This phenomenon hints at the vulnerability of the gut microbial environment after antibiotic treatment and emphasizes the critical role of microbial diversity in maintaining the integrity of the gut barrier.

In summary, HEE possesses the ability to modulate the composition of the gut microbiota by promoting the prevalence of beneficial genera.

We investigated the recovery after antibiotic treatment, the lipid-lowering effect of microorganisms, and the modulation of gut flora by HEE through a comprehensive analysis of the diversity of gut flora. This was achieved through a principal coordinates analysis (PCoA) of intestinal samples from various mouse groups (Figure 6A). The results revealed that NFNDC and NFHFD groups displayed slight separation along the PCO3 axis in PCoA, while NFHFDHEE showed significant divergence, indicating the efficacy of HEE in modulating gut microbiota diversity. Post-antibiotic treatment, the PGHFD group significantly diverged from the normal group along the PCO2 axis, indicating a significant change in the gut microbiota of the PG group due to the antibiotic. Principal component analysis (PCA) further highlighted distinctions between NFNDC and NFHFD groups along PC2, signifying the non-negligible effect of the HFD on the gut microbiota of mice (Figure 6B). Moreover, samples from the NFHFDHEE group showed differentiation from NFHFD along PC1, predicting the effects of the diet and HEE treatments on microbial community structure.

Finally, we utilized a heatmap for cluster analysis and inter-group differential comparison (Appendix A), identifying bacterial genera with relatively higher abundance in the intestinal microbiota of different mouse groups, aiming to discern patterns. In the heatmap, the central yellow section highlights significant bacterial genera in the PG group, including previously mentioned genera such as *Streptococcus*, *Stenotrophomonas*, *Enterobacter*, *Enterobacteriaceae*, and *Escherichia-Shigella*. These genera cluster together, forming a large group. Similarly, the elevated yellow section in the top-left corner represents the NF group post-HEE intervention, where genera like unidentified *Lachnospiraceae*, *Lactobacillus*, the *[Eubacterium] fissicatena* group, *Ruminiclostridium 9*, and *Romboutsia* also cluster into a large group. The distance between these two clusters is notably significant. The impacts of a high-fat diet, interventions by HEE, and antibiotic treatments have evidently contributed to the diverse composition of gut microbiota among different groups. HEE has been found to promote an increase in the abundance of beneficial microorganisms.

## 4. Discussion

The results for food intake and body weight (Figure 1) indicate that after 11–12 weeks of antibiotic treatment, the mice experienced a slowdown in weight gain, which then resumed after week 13 in the PGHFD group. This may be related to the use of high-dose quad antibiotics in the short term. Although there was a downward trend in weight gain, liver-to-body weight ratio, and blood glucose levels in the PGHFDHEE group, it was not significant. The difference between PGHFDHEE and NFHFDHEE was solely the involvement of the gut microbiota, revealing the potential impact of the gut microbiota on the weight loss and blood sugar reduction effects of HEE. Studies have shown a strong relationship between gut flora and obesity and blood sugar, and gut flora management has emerged as a new way to treat obesity [25]. Understanding the causal relationship between the gut microbiome and metabolic risk may help us to identify susceptible individuals for early, targeted intervention [26].

Liver and epididymal fat sections, along with liver function indicators (Figure 2), show that while HEE has a certain alleviating effect on the liver of HFD mice in the PG group, the effect is not significant, aligning closely with the observations mentioned earlier. Regarding lipid profiles and inflammatory markers, HEE contributes similarly across groups, effectively regulating blood lipids and suppressing inflammation to some extent (Figure 3). However, there are differences in emphasis between the NF and PG groups in terms of indicators; in the PG group, after HEE intervention, acetic acid levels rise while other fatty acid concentrations remain low, likely due to the antibiotic-induced depletion of the gut microbiota [27].

Several studies have reported that numerous acetic acid-producing bacteria have the potential to prevent the development of NAFLD [28]. In contrast, the NF group exhibited a decreasing trend in the levels of acetic and propionic acids in the cecum and colon, with concentrations in the cecum being 2–3 times higher than those in the colon. Additionally, significant changes in the levels of valeric acid and isovaleric acid were observed. In the cecum, the level of valeric acid was relatively high, and the promotional effect of HEE on valeric acid production could likely be due to an increased relative abundance of acid-producing bacterial populations. Studies have shown that *Lactobacillus acidophilus* can inhibit hepatocellular carcinoma associated with NAFLD by producing valeric acid [29].

In terms of regulating the gut microbiota, HEE adjusted the proportions of the *Bacteroidetes* and *Firmicutes* phyla and also modulated the abundance at the genus level (Figure 5 and Figure 6). It significantly increased the abundance of beneficial bacterial genera, including *Lactobacillus*, *Lachnospiraceae*, the *[Eubacterium] fissicatena* group, *Ruminiclostridium 9*, and *Romboutsia*. Conversely, some potentially harmful bacteria were also identified in the PG group, including *Streptococcus*, *Stenotrophomonas*, *Enterobacter*, *Enterobacteriaceae*, and *Escherichia-Shigella. Lactobacillus plantarum* offers protective benefits for HFD mice with NAFLD [30]. *Lactobacillus* rhamnosus regulates blood lipid levels and alleviates fatty liver by modulating a high-fat, high-cholesterol diet [31]. *Romboutsia* has the capability to modulate lipid metabolic functions in obese rats [32]. The *Lachnospiraceae* belongs to the phylum Firmicutes and is an important part of the intestinal microbial community. Members of this family are known for producing SCFAs, particularly butyrate and acetate, which are crucial for the host’s gut health and immune system function [33]. *Ruminiclostridium* species, including *Ruminiclostridium 9*, are recognized for their capacity to ferment dietary fibers and other indigestible carbohydrates, resulting in an increased production of intestinal SCFAs, such as butyrate, acetate, and propionate. In contrast, HEE had no significant effect on the intestinal flora of PG mice, which may be attributed to the strong destructive effect of antibiotics on the intestinal flora, masking the effect of HEE on the intestinal flora, and the harmful bacterial genera became dominant in the fragile intestines of the PG mice. A species within the genus *Streptococcus* has been reported as a potential risk factor for NAFLD. This indicates a significant link between certain bacterial species and the development or exacerbation of NAFLD [34]. *Stenotrophomonas* is a Gram-negative bacteria that includes a spectrum of species, from those commonly found in soil and plants to opportunistic human pathogens like *Stenotrophomonas maltophilia*. However, certain strains of *S. maltophilia* are known to be pathogenic to humans, presenting challenges due to their multidrug-resistant profiles [35]. *Enterobacter* can induce liver inflammation and exacerbate lipid accumulation [36,37]. At the same time, *Enterobacteriaceae* also acts as a trigger for endotoxins in NAFLD [38]. HEE interventions prevented the wild growth of these harmful genera, increased the abundance of beneficial genera, and increased the diversity of the flora. Overall, HEE modulates the intestinal flora, which translates into a pattern that benefits the alleviation of non-alcoholic fatty liver. Meanwhile HEE’s own effects still cannot be ignored. When the gut flora is disrupted, HEE is still able to alleviate NAFLD, although it is not as effective as when the gut flora is not disrupted.

## 5. Conclusions

In conclusion, our study demonstrated that HEE could effectively alleviate hepatic lipid accumulation, reduce blood lipids, and decrease the level of inflammation in both normal colony and pseudo sterile mice. HEE was able to regulate key enzymes involved in triglyceride catabolism to promote lipid metabolism, as well as inhibit the expression of fatty acid synthase and change the lipid metabolism pattern in the liver. Our study also highlights the effectiveness of HEE when the gut flora is disrupted and the potentiating effect of HEE modulation of the gut flora in normal flora for the treatment of NAFLD with HEE. HEE was found to affect the diversity of intestinal bacteria, improve the structure of the intestinal microbiota in mice, increase the level of beneficial bacterial species, and regulate the concentration of SCFAs in the intestinal tract. These beneficial effects were significantly reduced in PG mice. This study provides valuable insights into the efficacy and possible pathways of hawthorn in the treatment of NAFLD, which has a wide range of application scenarios as a common food ingredient. Future studies should focus on the exact components of hawthorn that alter the gut flora as well as the specific genera of gut flora with beneficial effects.

## Figures and Tables

**Figure 1 nutrients-16-01335-f001:**
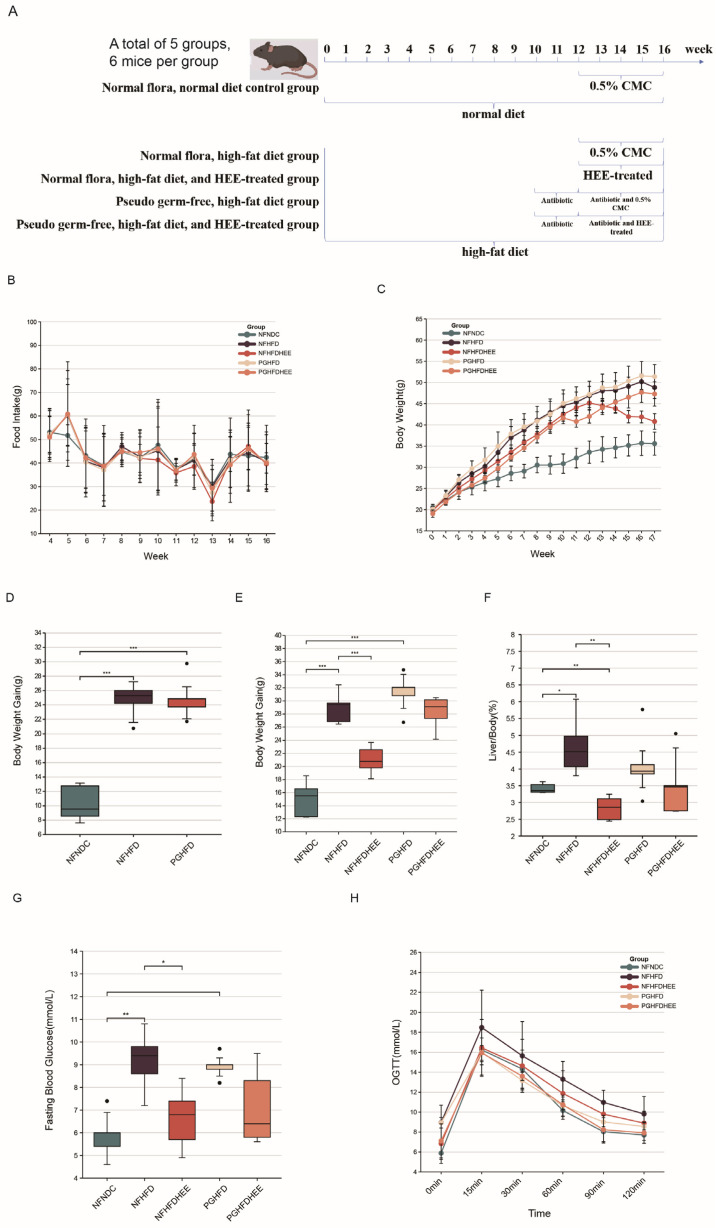
Effects of HEE on appetite, body weight, and blood glucose in high-fat-diet mice. (**A**) Animal experiment process. (**B**) Food intake in mice aged 4 to 16 weeks. (**C**) The body weights of mice in each group were recorded weekly from 0 to 16 weeks. (**D**) Establishment of a model for mouse food intake from 0 to 10 weeks. (**E**) Body weight gain during the intervention. (**F**) Liver-to-body weight ratio (100%). (**G**) Fasting blood glucose in mice at week 16. (**H**) Oral glucose tolerance of mice in each group. Data are shown as mean ± SD. (For each group, n = 5, * *p* < 0.05; ** *p* < 0.01; *** *p* < 0.001). The black dots in the graph indicate the values detected for each sample.

**Figure 2 nutrients-16-01335-f002:**
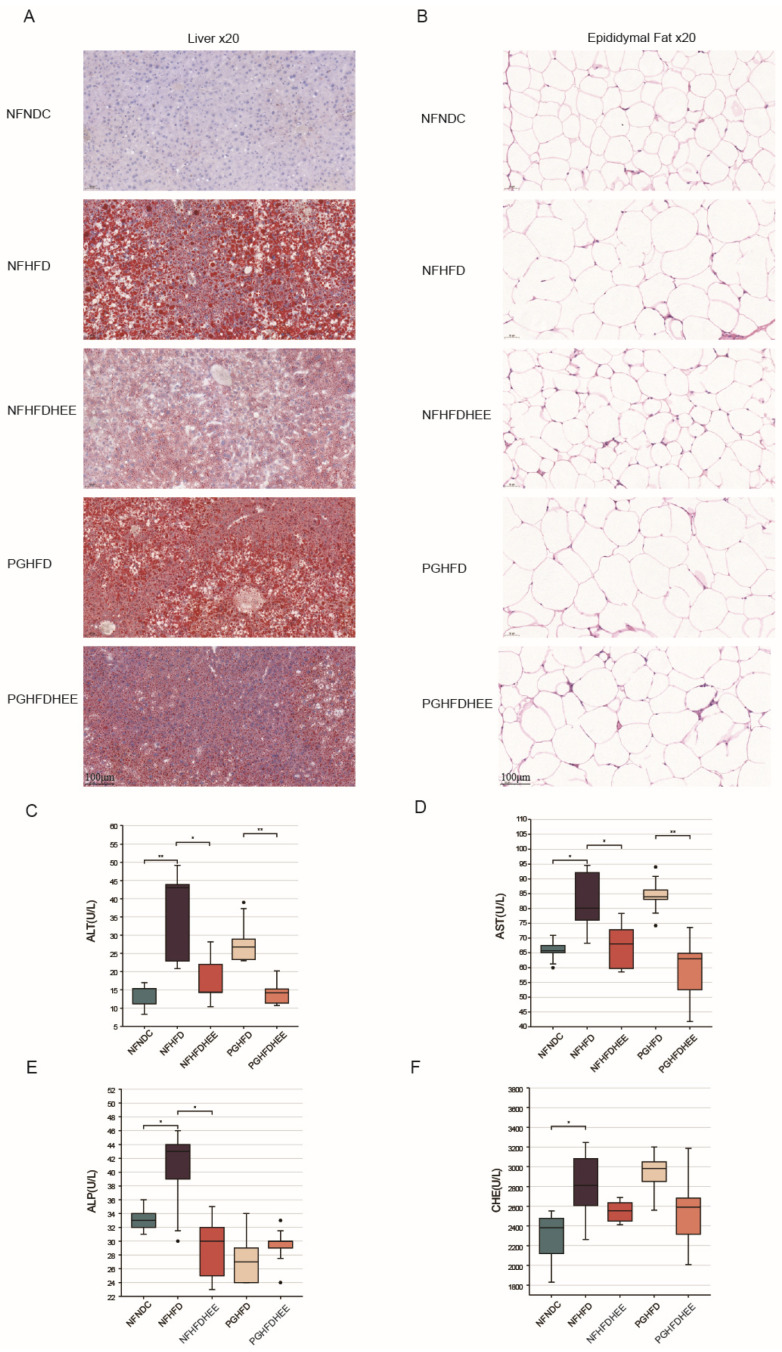
The effects of HEE on the liver function of high-fat-diet mice. (**A**) Oil red O-stained sections of the liver of mice in each group. (**B**) H&E-stained sections of epididymal fat from various groups of mice. (**C**–**F**) Serum biochemical indices in mice: alanine aminotransferase (ALT), glutamine aminotransferase (AST), alkaline phosphatase (ALP), and cholinesterase (CHE). Data are shown as mean ± SD. (For each group, n = 5, * *p* < 0.05; ** *p* < 0.01). The black dots in the graph indicate the values detected for each sample.

**Figure 3 nutrients-16-01335-f003:**
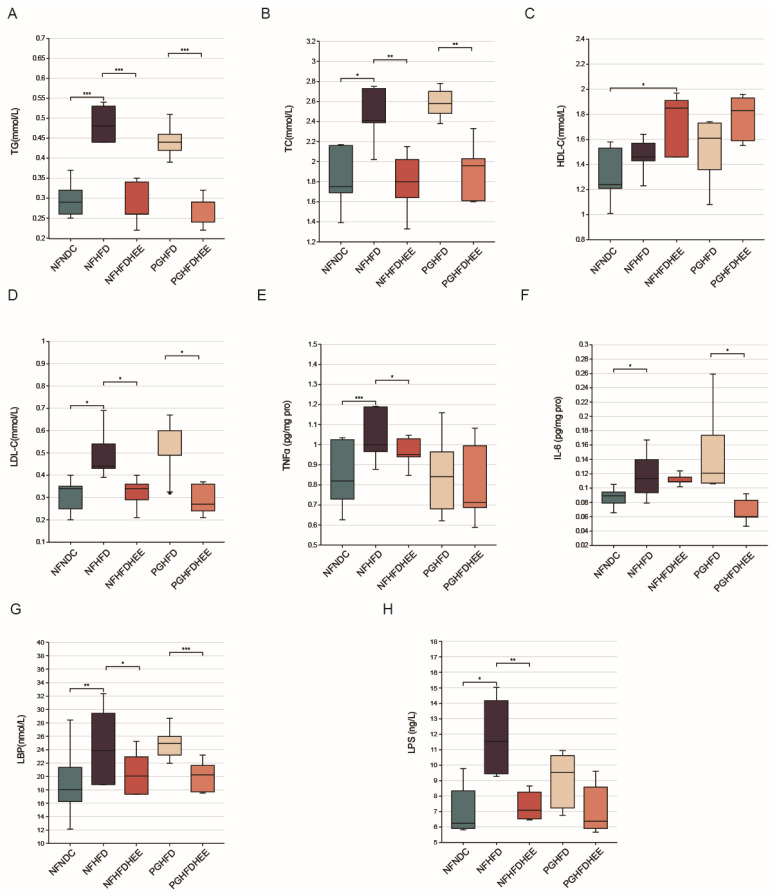
Effects of HEE on blood lipid levels and inflammation levels in mice. (**A**–**D**) Serum triglycerides (TG), serum total cholesterol (TC), high-density lipoprotein (HDL-c), and low-density lipoprotein (LDL-c) levels in all groups of mice. (**E**–**H**) Cytokine levels in mice of all groups: tumor necrosis factors α, interleukin-6, endotoxin lipopolysaccharide, and lipopolysaccharide binding protein. Data are shown as mean ± SD. (For each group, n = 5, * *p* < 0.05; ** *p* < 0.01; *** *p* < 0.001).

**Figure 4 nutrients-16-01335-f004:**
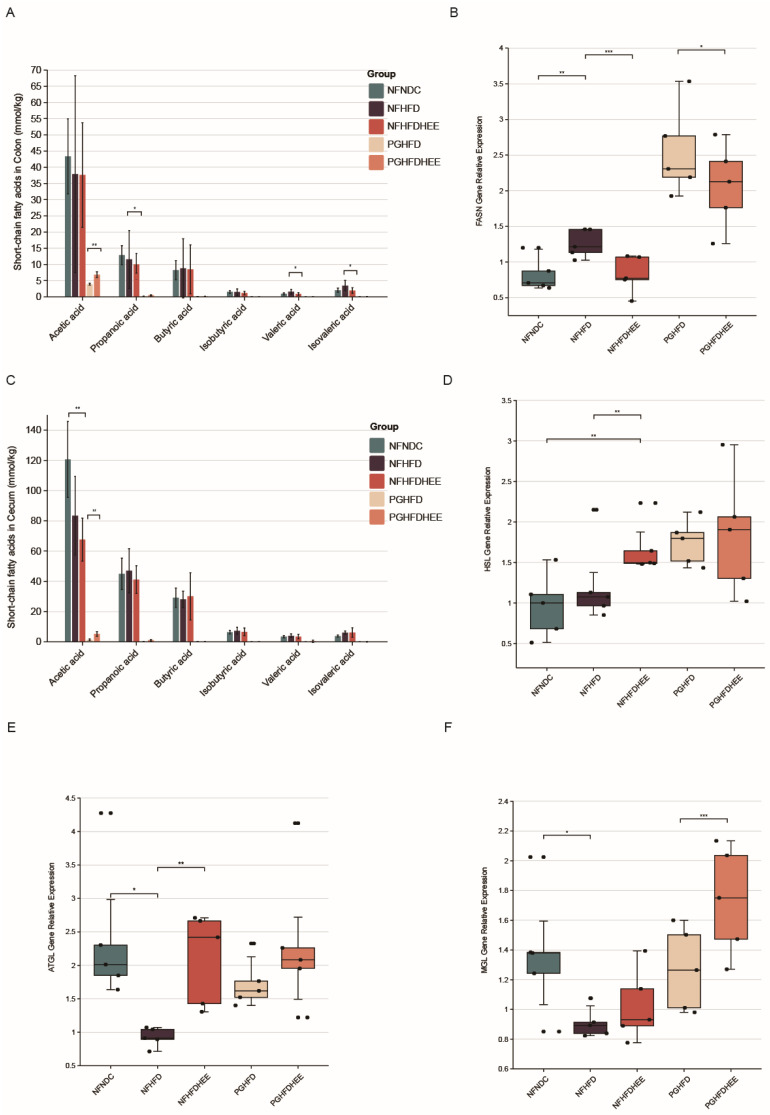
Regulation of short-chain fatty acids and hepatic lipid metabolism-related genes in the mouse intestine by HEE. (**A**) SCFAs in the colon. (**B**) FASN gene expression in liver. (**C**) SCFAs in the cecum. (**D**) HSL gene expression in liver. (**E**) ATGL gene expression in the liver. (**F**) MGL gene expression in the liver. Data are shown as mean ± SD. (For each group, n = 5, * *p* < 0.05; ** *p* < 0.01; *** *p* < 0.001). The black dots in the graph indicate the values detected for each sample.

**Figure 5 nutrients-16-01335-f005:**
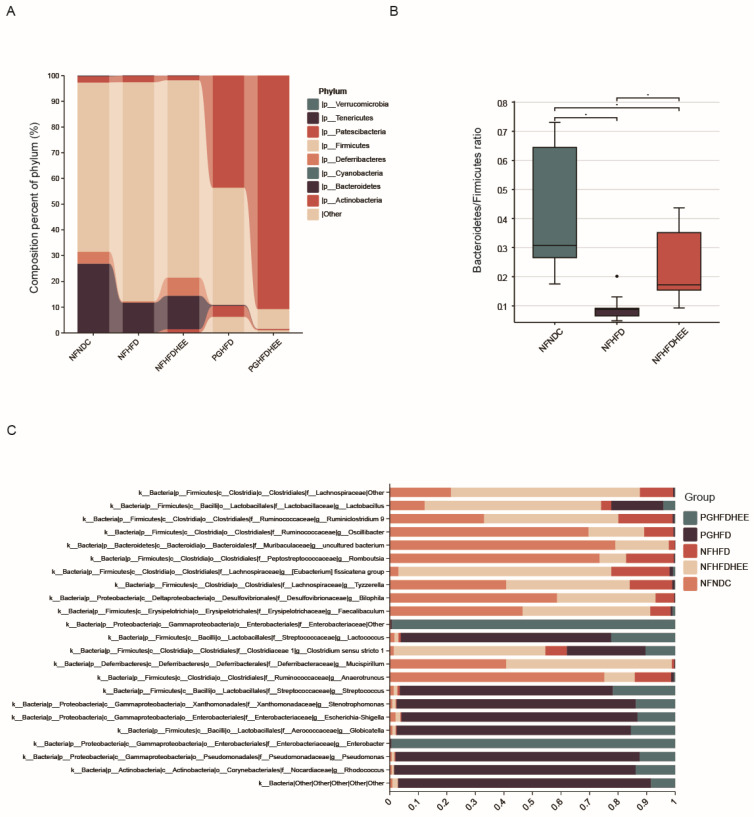
Regulatory effects of HEE on the gut microbiota of mice subjected to a high-fat diet. (**A**) The impact of HEE on the phylum level of gut microbiota in high-fat-diet mice. (**B**) The effects of HEE on *Bacteroidetes* and *Firmicutes* in the intestines. (**C**) Effect of HEE on the diversity of intestinal flora of mice at the genus level. Data are shown as mean ± SD. (For each group, n = 5, * *p* < 0.05).

**Figure 6 nutrients-16-01335-f006:**
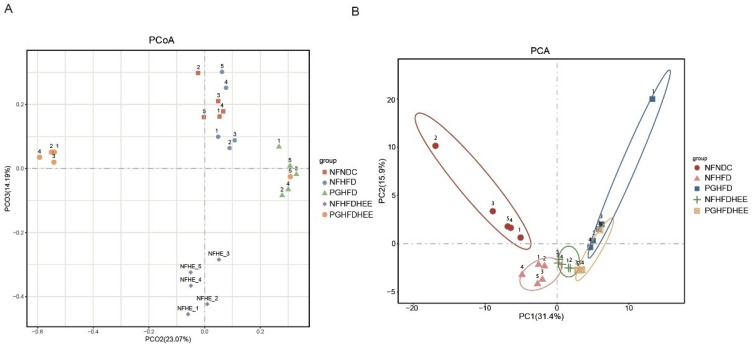
Modulation of intestinal species diversity in mice by HEE. (**A**) Principal co-ordinates analysis (PCoA) in the intestines of each group of mice. (**B**) PCA (principal component analysis) is applied in various groups of mouse intestines.

## Data Availability

All data presented in this study are available in the main body of the manuscript.

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
