# Peer review of "Targeting Non-Alcoholic Fatty Liver Disease with Hawthorn Ethanol Extract (HEE): A Comprehensive Examination of Hepatic Lipid Reduction and Gut Microbiota Modulation"

_nutrients, 2024, doi:10.3390/nu16091335_

Round 1

Reviewer 1 Report

Comments and Suggestions for Authors

Dear Authors.

Congratulations on such an interesting study assessing the effect of hawthorn extract on the animal model of NAFLD. The work is very well prepared in terms of content and its structure raises no objections. The topic of the work is fully implemented. I just have a question about what the term "certain food-medicine dual-purpose" means (in the introduction) (line 60-61).

Comments on the Quality of English Language

No comments.

Author Response

Thank you for your comments and kind advice.

"Certain dual-use substances" are substances that are recognized by the China Food and Drug Administration (FDA) as both food ingredients for use in a variety of foods and as medicines for the treatment of disease.

Reviewer 2 Report

Comments and Suggestions for Authors

The manuscript investigated the effects of HEE on NAFLD and the related gut microbiota modulation. The authors concluded that HEE suppressed the hepatic lipid accumulation by regulating the composition of gut microbiota. This work is interesting; however, the experiment design and the manuscript description should be improved.

Major concerns:

1. The similarity percentage is high (32%) and needs to be minimized.

2. A critical issue is the proper definition of HEE. Overall, the text, several definitions were found:

1) Title: Hawthorn Extract Enriched (HEE)

2) Line 98: Hawthorn Ethanol Extract (HEE)

3) Line 263, 305, 306: MFE??

4) Line 325: Holistic Extract Extract (HEE)

5) Line 359: high-efficiency ecosystems (HEE)

6) And numerous "Hawthorn extract" were used.

All abbreviations should be mentioned fully for the first time, and then they could be abbreviated.

3. Experiment design:

1) The NFC group also should be treated with 0.5% carboxymethyl cellulose to eliminate the effects of CMC;

2) Why did the authors define high-fat diet groups as "NFM" or "NFHE"? The readers may be confused by normal fat diet-treated conditions. The PGM and PGHE groups should also be revised as they were treated with a high-fat diet (HFD).

3) An experimental schema is needed for better understanding.

4) Line 110: "A dosage proven to have minimal side effects" => Who did this, when, or what side effects?

5) Section 2.3: The total experiment duration was 17 weeks. Ten weeks for HFD treatment, two weeks for antibiotic therapy, and four weeks for HEE administration, so what was the treatment in the final week?

4. The manuscript is wordy, with too much overlap and confusing expressions. For example:

1) Abstract: "HEE exhibits promise as a therapeutic agent for NAFLD management and gut health enhancement."

"The results propose that HEE could serve as a potential candidate for the prevention and treatment of NAFLD, offering novel perspectives on the impact of gut microbiota modulation on liver well-being." => Same meaning.

2) Line 162-163, 174-176: Unnecessary to provide well-known information.

3) Lines 289-290 should be deleted as it looks as a conclusion.

4) Line 368-374: Belong to Conclusion.

5) Line 376-385: Belong to Introduction.

5. The bioactive components of HEE should be listed.

6. So many carefulness mistakes:

1) Line 240: D) Food intake?

2) Line 77-78: Is it a title of literature?

3) The subtitles of 3.2 and 3.3 are different from the contents.

4) Figure 4B and 4C: Inconsistent description with the text.

5) Line 386: Did the antibiotic treat for 11 or 12 weeks?

7. Figure 2A and 2B: Please add the scale bar.

8. Figure 3E-3H: How to determine their levels? Please provide the primer sequences used for PCR or qPCR.

9. Figure 4A and 4B: Hard to recognize. The graphs should be revised. The description for these two graphs also should be polished for better understanding.

10. Figure 5A: Are there duplicate colors for different phyla?

   Figure 5B: Which time-point for measurement?

Minor concerns:

1. Again, all abbreviations should be mentioned fully for the first time. No full names were provided for NFM, NFHE, PGM, and PGHE.

2. Line 108: Please give the name of the high-fat diet.

3. 2.4: Please provide the paraffin's and frozen sections' thickness.

4. Change "rpm" to "g."

5. 2.7: Please provide all the reagent information.

6. Are there meaningful black dots in the graphs?

Comments on the Quality of English Language

Too many mistakes. An English language editing is recommended.

Round 2

Reviewer 2 Report

Comments and Suggestions for Authors

The authors revised the manuscript partially. They answered the questions from the reviewer without providing the related information in the manuscript. Some concerns remain and are needed to be improved.

1. Line 89-91: Please provide the percentage of the listed major components. What are the conceivable effective bioactive components contained in HEE?

2. The naming of the experiment group still needs to be more friendly to readers. The following are recommended:

Normal flora, normal diet control group: NFNDC

Normal flora model, high-fat diet group: NFMHFD

Normal flora model, high-fat diet, and HEE-treated group: NFMHFDHEE

Pseudo germ-free model, high-fat diet group: PGMHFD

Pseudo germ-free model, high-fat diet, and HEE-treated group: PGMHFDHEE

3. The reviewer didn’t find the graphical abstracts. The authors are strongly recommended to add the experimental schema as a figure, not a graphical abstract.

4. Line 130: “eosin”, not “eosion.”

5. Line 132: “Frozen sections were then cut into 10-μm thin slices at low temperature and stained with oil red O and H&E to assess lipid accumulation.” -> There is no eosin staining in the protocol for Oil red O staining, only hematoxylin staining.

6. The methods for measuring TNF-α, IL-6, LBP, and LPS are missing. The authors answered the reviewer's question but also needed to add the information to the manuscript.

7. Again, please provide the primer sequences or TaqMan probes used for PCR or qPCR.

8. Figure 1A and 1F: “Food intake in mice aged 1 to 13 weeks” and “Fasting blood glucose in mice at week 18” should be revised.

9. Figure 4A and 4C:

1) Different legends from the related graphs.

2) In the text, “concentration of SCFAs dropped significantly after antibiotic treatment, highlighting the crucial role of the gut microbiome (Figure 4A, B).” However, the graph of Figure 4B shows FASN gene expression in the liver.

10. Figure 5B: The authors answered that the fecal samples were collected the day before the mice were sacrificed. Please provide the information in the M&M section.

11. Line 315-317: “Post-antibiotic treatment, the PGM group significantly diverged from the normal (NF) group along the PCO2 axis, indicating a significant change in the gut microbiota of the PG group due to HEE.” => Confused description. How do you explain the significant difference between PGHE_5 and the other four mice?

Comments on the Quality of English Language

Small mistakes are found. Please check the entire manuscript.

Round 3

Reviewer 2 Report

Comments and Suggestions for Authors

Two minor concens:

1. The experimental schema should be numbered as Figure 1A.

2. Answer to Comment 7: Does Table 2 (line 202) mean Supplementary Table 2?
